# Evaluating the Effect of Hypoxia on Human Adult Mesenchymal Stromal Cell Chondrogenesis In Vitro*:* A Systematic Review

**DOI:** 10.3390/ijms232315210

**Published:** 2022-12-02

**Authors:** Charindu K. I. Ranmuthu, Chanuka D. S. Ranmuthu, Chalukya K. Wijewardena, Matthew K. T. Seah, Wasim S. Khan

**Affiliations:** Division of Trauma and Orthopaedics, Addenbrooke’s Hospital, Cambridge CB2 0QQ, UK

**Keywords:** mesenchymal stromal cells, chondrogenesis, hypoxia

## Abstract

Human adult mesenchymal stromal cells (MSCs) from a variety of sources may be used to repair defects in articular cartilage by inducing them into chondrogenic differentiation. The conditions in which optimal chondrogenic differentiation takes place are an area of interest in the field of tissue engineering. Chondrocytes exist in vivo in a normally hypoxic environment and thus it has been suggested that exposing MSCs to hypoxia may also contribute to a beneficial effect on their differentiation. There are two main stages in which MSCs can be exposed to hypoxia, the expansion phase when cells are cultured, and the differentiation phase when cells are induced with a chondrogenic medium. This systematic review sought to explore the effect of hypoxia at these two stages on human adult MSC chondrogenesis in vitro. A literature search was performed on PubMed, EMBASE, Medline via Ovid, and Cochrane, and 24 studies were ultimately included. The majority of these studies showed that hypoxia during the expansion phase or the differentiation phase enhances at least some markers of chondrogenic differentiation in adult MSCs. These results were not always demonstrated at the protein level and there were also conflicting reports. Studies evaluating continuous exposure to hypoxia during the expansion and differentiation phases also had mixed results. These inconsistent results can be explained by the heterogeneity of studies, including factors such as different sources of MSCs used, donor variability, level of hypoxia used in each study, time exposed to hypoxia, and differences in culture methodology.

## 1. Introduction

Articular cartilage is found at the ends of long bones and acts to minimise joint friction, thereby improving the ease of movement at these joints. Cartilage has a low intrinsic capacity for repair, and damage to the articular cartilage may lead to progressive damage to the joint [1]. Cartilage consists of cells called chondrocytes which are surrounded by an extracellular matrix made of collagen, proteoglycans, water, and noncollagen proteins [2]. Type II collagen is the most abundant collagen type, and it polymerizes with the other main collagen types, type IX and XI, to produce a hetero fibril network [3]. Other minor collagens include types III, IV, V, VI, X, XII, XIV, and XXVII [2]. The collagen fibrils interact with other matrix proteins, the most abundant being aggrecan which forms aggregates with hyaluronic acid [4]. Osteoarthritis (OA) is a disease characterised by degeneration of the articular cartilage, leading to joint stiffness and pain [1,5]. In OA cartilage, the main collagens change from type II collagen into mixtures of types I, II, and III collagen [3].

The changes associated with progressive cartilage wear can result from chronic changes in extracellular matrix regulation, which is produced and regulated by chondrocytes [6]. Chondrocyte hypertrophy and proteolytic enzymes are implicated in this process. Type X collagen is produced by hypertrophic chondrocytes and is only usually expressed in OA cartilage [7]. It acts to regulate the metabolism of chondrocytes and ensure tissue stiffness. Proteolytic enzymes such as matrix metalloproteases (MMPs) are involved in the turnover of the extracellular matrix. MMPs 1 and 13 catabolise the collagen framework while MMP3 and ADAMTS-4 (A disintegrin and metalloproteinase with thrombospondin motifs 4) cause proteoglycan breakdown through the upregulation of inflammatory cytokines such as interleukin (IL)-1β, tumour necrosis factor-alpha (TNF-α), and IL-6 [8,9,10].

Existing clinical treatments for OA in the form of chondrocyte implantation, microfracture, and osteochondral transplantation may only delay degradation and thus new therapeutic strategies are needed [11]. MSCs are multipotent cells that can differentiate into cartilage progenitor cells, making them a prime candidate in cartilage regenerative therapies. For their standardised use, MSCs must be characterised and the International Society for Cellular Therapy has published a consensus minimal criteria for their definition. This includes the positive expression of the cell surface markers CD105, CD73, and CD90 and the negative expression of CD45, CD14, CD34, CD19, and human leukocyte antigen (HLA) Class II. Additional criteria include the ability to adhere to plastic and the capacity for trilineage differentiation (into osteoblasts, adipocytes, and chondroblasts).

MSCs can be isolated from several sources, such as bone marrow, adipose tissue, umbilical cord blood, and synovial fluid [12]. Along with their relative ease of isolation, their ability to differentiate into chondrocyte progenitors as well as their immunomodulatory properties [13] make them promising candidates for cartilage regenerative therapies.

There are two main methods for the three-dimensional (3D) culture of MSCs: utilising scaffolds, pellet, or micromass culture [14]. The use of scaffolds has previously been shown to improve chondrogenic differentiation [15]. Combining MSCs with chondrogenic factors such as growth factors (e.g., transforming growth factor beta (TGF-β), bone morphogenetic protein 2 (BMP2), insulin-like growth factor (IGF)), platelet-rich plasma, or engineered scaffolds, have also been shown to promote the migration and differentiation of MSCs as well as supplying sufficient mechanical integrity for successful surgical implantation [16,17].

The culture and expansion of MSCs in vitro for their therapeutic use in diseases such as OA require an efficient process whereby specific conditions are used to maximise chondrogenic potential. Cartilage is an avascular tissue where inner layer chondrocytes are usually found in a hypoxic (1–10% O_2_) environment. Some studies have therefore examined the beneficial effects of culturing MSCs in a similar environment in vitro in order to better replicate the conditions in vivo [18,19,20].

MSCs can be exposed to hypoxic culture in either or both of these two protocol stages: the expansion stage or the differentiation stage. Previous systematic reviews have included both human-derived and animal-derived MSCs, but this review seeks to summarise the current research on the effect of hypoxia on the chondrogenesis of human adult MSCs in vitro.

## 2. Results

Twenty-four studies met the criteria for inclusion and were included in this review (Table 1). Information such as the tissue source of the MSCs, cell surface markers assessed using flow cytometry, whether a pellet/scaffold/monolayer culture system was used, the oxygen tension used, and the stage(s) at which the groups were exposed to hypoxia are presented. Table 2 summarises the outcome measures used in each study and the results of the effect of hypoxia on chondrogenesis in each study.

### 2.1. Tissue Source of MSCs Used

The majority of the studies (15/24) used bone marrow derived MSCs. Six out of 24 studies used adipose tissue and 2/24 studies used synovium derived MSCs. One out of 24 studies used both human bone marrow and human adipose tissue [26]. Wang et al., 2021 [26] also included rat bone marrow and adipose tissue derived MSCs, but these groups were excluded from this review [26].

### 2.2. MSC Markers Assessed

A range of MSC markers was used to characterise the cells as detailed in Table 1. The most common markers assessed across studies were CD105 (20/24 studies) and CD90 (20/24 studies).

### 2.3. Stage of Exposure to Hypoxia

Five out of 24 studies exposed MSCs to hypoxia during the expansion phase only [26,30,31,37,44] and 12/24 studies exposed MSCs during the differentiation phase only [22,24,25,27,28,29,32,34,35,36,40,41]. In one study, Van de Walle et al. [41] exposed MSCs to hypoxia and then normoxia in the differentiation phase and compared this to continuous hypoxic culture [41]. Four out of 24 studies contained two groups where MSCs were exposed to hypoxia continuously both during the expansion and the differentiation phase compared to a group exposed to normoxia during both the expansion and the differentiation phase [38,39,42,43].

There were three other studies that studied sequential exposure of hypoxia in the expansion phase followed by the differentiation phase [21,23,33]. Adesida et al. [21] and Boyette et al. [23] both contained four groups that exposed stem cells to (1) hypoxia during expansion with normoxia during differentiation, (2) hypoxia during expansion with hypoxia during differentiation, (3) normoxia during expansion with normoxia during differentiation, and (4) normoxia during expansion with hypoxia during differentiation [21,23]. Markway et al. [33] exposed both experimental groups to hypoxia during the expansion phase and then exposed cells to either hypoxia during the differentiation phase or normoxia during the differentiation phase [33].

### 2.4. Effect of Hypoxia on Chondrogenic Differentiation: Expansion Stage Only

The majority of studies in this category (3/5 studies) reported that the potential for chondrogenic differentiation was enhanced with exposure to hypoxia during the expansion stage only [26,30,31]. One out of five studies showed no difference following hypoxic or normoxic cultures during the expansion stage [37] but 1/5 of studies showed the opposite effect, where cells cultured under normoxia during the expansion phase showed significantly greater expression of Sox9 (transcription factor essentially during chondrogenesis) compared to cells cultured under hypoxic conditions during the expansion phase [44].

### 2.5. Effect of Hypoxia on Chondrogenic Differentiation: Differentiation Stage Only

The majority of studies (11/12) showed that hypoxia during the differentiation phase only may enhance some markers of chondrogenic differentiation [22,25,27,28,29,32,34,35,36,40,41]. There were studies within this that showed some markers of chondrogenic differentiation were not enhanced under hypoxia [28,34,35,36]. Notably, Neybecker et al. [36] showed that markers of chondrogenic differentiation were upregulated at the gene level but not at the protein level [36]. Van de Walle et al. [41] interestingly exposed stem cells to either hypoxia continuously during the differentiation phase or to normoxia followed by hypoxia during the differentiation phase [41]. They showed that the latter may be associated with increased chondrogenesis. Cicione et al. [24] demonstrated that in severe hypoxia (1%) during the differentiation phase, chondrogenic differentiation may be inhibited compared to normoxic conditions [24].

### 2.6. Effect of Hypoxia on Chondrogenic Differentiation: During Both Expansion and Differentiation Phase

There were mixed conclusions from the four studies that exposed cells to continuous hypoxic or normoxic culture. Pattappa et al. [38] showed that there were two groups of donors: either hypoxia responsive or unresponsive. For those that were responsive, continuous hypoxia enhanced the expression of chondrogenic markers whilst for the unresponsive group, there was no change observed [38]. Pattappa et al. [39] also showed no difference in chondrogenic markers [39]. Safwani et al. [42] showed normoxia actually produced the best chondrogenic outcomes when comparing normoxia without fetal bovine serum (FBS) to all other groups in their study (including hypoxia without FBS) [42]. In contrast, Weijers et al. [43] showed hypoxia continuously enhanced chondrogenic markers [43].

### 2.7. Effect of Sequential Exposure to Hypoxia in Both the Expansion Phase and Then the Differentiation Phase

Adesida et al. [21] investigated all combinations of hypoxia/normoxia during the expansion and differentiation phases [21]. They showed that hypoxia during the expansion phase may enhance chondrogenic differentiation irrespective of the oxygen tension used during the differentiation phase. In contrast to this, Boyette et al. [23] also conducted a similar experiment but concluded that chondrogenesis is enhanced when cells are exposed to normoxia in the expansion phase and then hypoxia in the differentiation phase [23]. Markway et al. [33] exposed both groups to hypoxia during the expansion phase and then exposed one group to hypoxia during differentiation and the other to normoxia during differentiation. They demonstrated that some markers of chondrogenic differentiation were more significantly expressed following exposure to hypoxia during the differentiation phase [33].

## 3. Discussion

There were several limitations to this review. Four databases were searched and only studies in English were screened. Although included studies attempted to characterise MSC cell surface marker expression using flow cytometry, not all studies used the minimum criteria as outlined by ISCT criteria [45]. As studies in the literature on the effect of hypoxia on MSC chondrogenesis use variable isolation and expansion protocols, it is difficult to make comparisons between studies.

Many of the studies we investigated also commented on collagen X expression [26,32,35,41]. As shown by Markway et al. [33] an effect of hypoxic culture is associated with enhanced expression of *COL10A1* (required for Type X collagen X), which is also a marker of hypertrophic differentiation [33]. They suggested that a plasma polymer coating or the use of chondroitin sulphate and parathyroid hormone-related protein (PTH-RP) may be useful in minimising this. Additionally, Van de Walle et al. [41] comment on how chondrogenesis may need to be stopped at specific time points to limit this hypertrophy, such as stopping chondrogenesis at day 14 [41]. Further investigation into these time points should take place to determine the optimum length of time to expose different groups in order to limit hypertrophy. Another limitation is the use of real-time quantitative polymerase chain reaction (RT-qPCR) to quantify gene expression in the majority of studies. While this provides important transcriptomic information; this does not necessarily translate to expression at the proteomic level. All the studies used histological methods to identify changes in protein expression or tissue composition (e.g., the presence of Type II collagen) but changes were not quantified.

Hypoxic culture during the expansion phase only may act to enhance some markers of chondrogenic differentiation as seen in the majority of studies included in this review. There were, however, two conflicting reports, O’Hara et al. [37] and Zhao et al. [44] which showed no difference following hypoxic culture, and normoxia enhancing chondrogenesis respectively [37,44]. Hypoxia during the differentiation phase may also enhance at least some markers of chondrogenic differentiation as shown in the majority of these studies. However, it is important to note that these changes were not always demonstrated at the protein level [36] nor in all markers [28,34,35]. One conflicting report, by Cicione et al. [24] showed severe hypoxia during the differentiation phase inhibited chondrogenic differentiation, compared to normoxia [24]. The results of continuous exposure to hypoxia during both the expansion and differentiation phases were mixed, as described in the results section.

The variability of these results may be explained by variations in the methodology of these studies, including different sources of MSC, MSC donor variability, different levels of hypoxia, differences in time of exposure in experiments to hypoxia, differences in growth factors/additive factors used in the culture medium, and studies using scaffold or pellets or monolayer culture.

### 3.1. Donor Variability of MSCs

As discussed by Olmedo-Moreno et al. [46] the heterogeneity in the donor age and co-morbidities can influence the chondrogenic potential of the MSCs [46]. This heterogeneity may be compounded by the differences in protocols used to isolate and expand the MSCs. This was demonstrated in bone MSCs by Stroncek et al. [47] who obtained these cells from the same source and showed that labs using different propagation methods produced MSCs with different functional abilities including differences in gene expression [47].

Also, MSCs are intrinsically a heterogenous population of cells, and Pattappa et al. [38] showed this by demonstrating both responsive and nonresponsive MSC populations to hypoxia within their donor groups [38].

Within this review, there were variations between studies in whether they used diseased or healthy donor MSCs. For example, Legendre et al. [32] used diseased donors whereas others such as Markway et al. [33] used healthy donors as the source of bone marrow stem cells. Indeed, as discussed by Legendre et al. [32] the effect of using healthy compared to disease donors on chondrogenic differentiation is not fully understood and warrants further investigation [32]. Some studies have reported that MSCs derived from OA donors compared to healthy donors showed a reduced capacity for chondrogenic differentiation [48].

### 3.2. Different Stage and Time of Exposure in Experiments to Hypoxia

Of the studies included in this review, each study exposed cells to hypoxia at different stages in the process. Of the two studies that compared all combinations of hypoxia/normoxia during both the expansion phase and the differentiation phase, both authors reported different outcomes [21,23]. Adesida et al. [21] concluded that hypoxia during the expansion phase enhances chondrogenic differentiation whereas Boyette et al. [23] concluded that chondrogenesis was best when cells were exposed to normoxia in the expansion phase and then hypoxia [21,23]. Again, the other confounding variables described in this section may contribute to this. Moreover, each study included in this review exposed cells to hypoxia for different lengths of time further confounding the results.

### 3.3. Differing Conditions Used in Both Culture Stage and Differentiation Stage

The culture protocols, including the additive factors such as growth factors used either during the expansion or differentiation phases, vary between studies. The influence of culture conditions has been discussed previously, in ref. [49]. Wan Safwani et al. [42] show that adipose-derived MSCs may be best cultured in normoxia without serum compared to groups such as normoxia with serum and hypoxia without serum [42].

The influence of additional factors during the differentiation phase has also been evaluated. For example, Legendere et al. [32] showed that the addition of factors such as transforming growth factor-beta1 (TGF-β1) and bone morphogenetic protein-2 (BMP-2) may enhance the effect of hypoxia on chondrogenic differentiation [32]. However, other studies used no additional growth factors to enhance chondrogenic differentiation [25].

To facilitate the optimum chondrogenic differentiation of MSCs, the optimum culture conditions including the culture media, growth factors, and hypoxia should be established. However, it is important to note that the effect of hypoxic culture may also change with different culture media and growth factors.

### 3.4. Different Characterisation of MSCs

Despite the presence of a minimum criteria for the characterisation of MSCs [45], the cell markers chosen for this characterisation can be different in various studies [50], including in clinical trials [51]. In addition to this, the nomenclature used in studies requires more clarification. Studies in the literature confusingly use the term MSC to mean “mesenchymal stem cells” and also “mesenchymal stromal cells”. As discussed by Horwitz et al. [52] the term MSC should only be used when cells meet certain criteria [52]. The minimum criteria [45] include the positive expression of CD105, CD73, and CD90 and it is therefore no surprise that the majority of studies use these markers to define their MSCs (Table 3). As an example of the variety of additional markers used, one study used Stro1 as a marker for MSCs, although Stro1 is also highly expressed in endothelial tissues; one study used stage-specific embryonic antigen-4 (SSEA-4), a marker also expressed in embryonic stem cells, and one study used CD140b (also known as platelet-derived growth factor receptor-β) which is additionally expressed in smooth muscle and glial cells.

### 3.5. Risk of Bias

A number of possible tools to evaluate the risk of bias were assessed for suitability in this review but none were deemed totally applicable to in vitro studies. However, certain themes emerge which may help evaluate these studies. For example, the use of a bioreactor as used in Lim et al. [27] may be beneficial to reduce the potential for confounding variables in experiments such as pH, temperature, and pressure can be controlled, thus improving the internal validity for results.

### 3.6. Possible Biochemical Mechanisms of the Effect of Hypoxia on Chondrogenesis

It has long been established that the oxygen tension level in articular cartilage is below 5% and studies have shown that partial pressure of oxygen (pO2) levels in the human bone marrow is around 51.8 mmHg [53,54]. There have been several suggestions in the literature as to how hypoxia may influence chondrogenesis at the gene level. Hypoxia-inducible factor (HIF)-1α and HIF-2α are two transcription factors that have been implicated. HIF-1α and HIF-2α are hydroxylated by specific prolyl hydroxylase domain (PHD) proteins, which require oxygen as a cofactor. This leads to the binding of von Hippel–Lindau protein (pVHL) and then its subsequent proteasomal degradation [55]. In hypoxic conditions, the activity of PHDs is inhibited leading to the stabilisation of HIF-1α and HIF-2α, enabling its heterodimerization with the HIF- β subunit which then activates target genes.

HIF-1α has a key role in both the differentiation of progenitor cells as well as maintaining a chondrogenic phenotype once differentiated. For example, during development, Duval et al. [25] suggest that HIF-1α causes an increase in *SRY-box transcription factor 9* (*Sox9*) expression. SOX9 activates the enhancer elements in genes such as *Col2a1* (coding for Type II collagen) and *ACAN* (coding for aggrecan) thereby promoting the secretion of extracellular matrix components [56,57]. HIF-1α also has been suggested to have additional influences on cellular responses to hypoxia, such as increasing markers such as Octamer-binding transcription factor four (Oct-4), reduced expression-1 (Rex-1), human telomerase reverse transcriptase (hTeRT), and SSEA-4 expression [58]. Hypoxia is thought to cause HIF-1α accumulation in the nucleus which forms a complex with HIF-1β. In higher oxygen concentrations, HIF-1α is degraded leading to a promotion of a hypertrophic/osteoblastic lineage [59]. This results in the suppression of mitochondrial respiration due to a switch between glycolysis and oxidative phosphorylation. In this way, HIF-1α regulates anaerobic respiration at these different oxygen concentrations [60]. Once developed, HIF-1α is also implicated in reducing osteogenesis and chondrocyte hypertrophy through reducing expression of hypertrophic markers like *COL1A1* (coding for Type I collagen) and *COL3A1* (coding for Type III collagen) [59]. Protecting chondrocytes from apoptosis- knockouts of HIF-1α in OA models have been shown to cause an increase in apoptosis of chondrocytes [61]. HIF-1α may act to reduce the expression of catabolic enzymes such as MMP-13, which act to degrade the articular cartilage thereby preventing its destruction [62].

The role of HIF-2α has also been studied and is much debated. Initial evidence suggested that there are distinct roles for HIF-1α and HIF-2α. Whereas HIF-1α may be more involved in the production and maintenance of articular cartilage, HIF-2α was implicated more in the regulation of ossification and cartilage destruction by its promoter action on catabolic genes [63]. However, other evidence suggests that HIF-2α may also promote the expression of genes involved in cartilage matrix secretion [27]. The effect of hypoxia on HIF-1α, and HIF-2α is also a matter of debate. Adesida et al. [21] found that hypoxia promoted the mRNA expression of HIF-2α but found no such effect on HIF-1α [21]. They, therefore, suggested that the transcriptional activity of HIF-2α acts as a mediator for the response of MSCs to a low oxygen tension environment. Khan et al. [28] similarly suggested that HIF-2α, but not HIF-1α, activity is enhanced. They went on to propose that HIF-2α may increase the expression of transcription factors such as SOX9, SRY-box transcription factor five (SOX5), and SRY-box transcription factor six (SOX6), which could then indirectly increase cartilage matrix gene expression through both SOX9 and other SOX9-independent mechanisms [28]. There remains no consensus, as Lee et al. [30,31] reported, that hypoxia does indeed activate the HIF-1α pathway [30]. Several differences between these studies must be noted. First, they used MSCs derived from different tissue sources (Khan et al. [28] used adipose-derived MSCs, and Lee et al. [30] used bone marrow). Further, Khan et al. [28] exposed cells to hypoxia during the differentiation phase. This was in contrast to Lee et al. [30] who exposed the MSCs to hypoxia during the expansion phase. 

### 3.7. The Effect of Hypoxia on Chondrocyte Maturation and Function

There has also been evidence that exposure to hypoxia may affect chondrocyte maturation and function. For example, in a study using C3H10T1/2 cells, a cell line derived from mouse embryo cells, hypoxia in the presence of BMP-2 decreased *COL10A1* (coding for type X collagen) gene expression and thus was seen to downregulate the terminal differentiation of chondrocytes in endochondral ossification. Ref [64] BMP-2 is thought to be heavily involved in the induction of chondrocyte differentiation [65]. It was suggested that hypoxia, therefore, acts to prevent hypertrophy and maintain the chondrocyte phenotype. The mechanism for this is suggested by Hirao et al. [64] in which hypoxia causes a reduction in *COL10A1* expression through three main pathways: increasing BMP-2-induced activation of p38 mitogen-activated protein kinases (p38 MAPK), downregulation of the Smad pathway, and upregulation of the *HDAC2* gene which all culminates in decreased expression of *Runt-related transcription factor two* (*Runx2*), and in turn acts to downregulate *COL10A1* expression. Their findings, however, suggested that hypoxia stimulated chondrocyte differentiation through Sox-9 independent pathways, contrary to the findings of Duval et al. [25]. This study however was performed in mouse mesenchymal cells rather than human adult MSCs, as focused on in this review. This illustrates the need for further work into the mechanism of hypoxia on chondrocyte differentiation, maturation, and function in human adult MSCs.

Hypoxia has been shown to have significant effects on MSC function. Ejtedhadifar et al. [19] demonstrated that MSCs exposed to hypoxia demonstrate altered gene expression in relation to metabolism. For example, hypoxia is associated with the upregulation of genes such as *pyruvate dehydrogenase kinase-1* (*PDK-1*) which acts to decrease the use of pyruvate in the Krebs cycle [19]. Moreover, hypoxia has been shown to increase intracellular calcium in MSCs—this is a key ion in physiological function, particularly involved in chondrocyte mechano-transduction and regulatory volume changes [66]. With the number of key players in cartilage regeneration, the effects of hypoxia on the various cell types (ranging from progenitor cells to differentiated cells) should be investigated in more detail.

### 3.8. Future Implications

MSCs must be defined correctly and characterised according to the ISCT minimum criteria [45]. Optimum culture conditions including growth factors, growth medium, level of oxygen tension, and optimum stages for hypoxic culture also need to be established in vitro. As well as environmental factors, biomechanical factors may also play a part and could be optimised. For example, Ge et al. [67] showed that dynamic compression loading may promote the chondrogenic differentiation of MSCs derived from the synovium [67]. In order to facilitate comparison between studies, bioreactors such as those used by Lim et al. [27] could be utilised in a closely monitored environment and reduce confounding variables [27]. These bioreactors can also be used to study the effect of biomechanical loading on MSCs [68]. A further way to enable comparison between studies is to standardise the quantification of chondrogenesis, due to observed differences in both gene expression and protein translation. To encourage the translation of this knowledge into clinical therapies, it will be important to investigate the effect of hypoxia in MSCs when they are transplanted into preclinical models. Indeed, initial studies have been performed, including some studies included in this review. Duval et al. [25] demonstrated that human bone marrow-derived MSCs, cultured in hypoxia compared to normoxia, and grown on alginate beads before implantation into mice, appeared to produce a matrix that was more similar to hyaline cartilage than the normoxia group [25]. Further work will need to be done to see if these results can be translated into clinically relevant outcomes.

## 4. Methods

A literature search was performed in July 2022 using PubMed, EMBASE, Medline via Ovid, and Cochrane. The search terms were: “mesenchymal cell”, “hypoxia or physoxia”, and “chondrogenesis or chondrogenic”. A hand search of review articles related to the subject was also performed for any other articles that may fit the inclusion and exclusion criteria. The inclusion criteria were studies that focused on the effect of hypoxia compared to normoxia (exposure should be more than 48 h); studies relevant to articular cartilage chondrogenesis; and studies using only adult MSCs. Studies had to characterise MSCs with common MSC markers, which included the positive expression of the cell surface markers CD13, CD29, CD44, CD73, CD90, CD105, CD106, CD146, CD151, or CD166 and the negative expression of CD11b, CD14, CD19, CD31, CD34, CD45, or CD64.

The exclusion criteria were studies that used mimetic agents to mimic inducing a hypoxic environment; all review and conference articles; animal studies; articles for which no free full text was available; and those not published in the English language.

Two reviewers (CKIR and CDSR) initially independently performed abstract screening with the aforementioned inclusion and exclusion criteria. Those chosen for full-text analysis were then also independently assessed by these two reviewers. Disputes were resolved by a third reviewer (WSK). Twenty-four studies were ultimately chosen for this review. A flow diagram of this process is displayed in Figure 1.

## Figures and Tables

**Figure 1 ijms-23-15210-f001:**
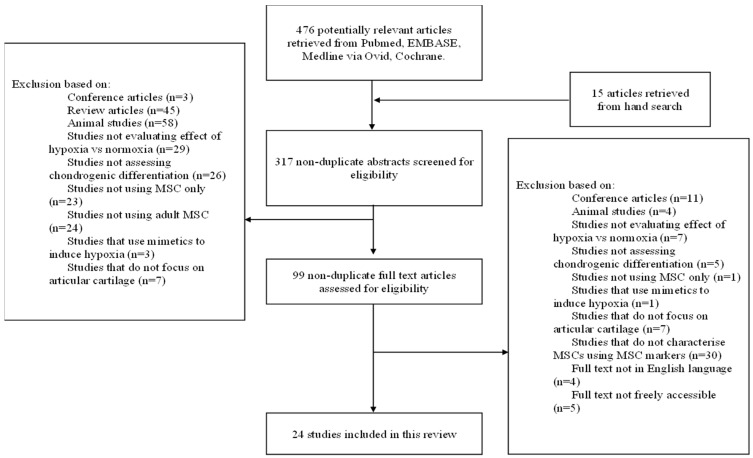
Flow chart showing the process of study selection for inclusion in this systematic review.

**Table 1 ijms-23-15210-t001:** Study Characteristics. Shows the following information for each study included in this review: source of MCSs, MSC markers assessed for, pellet or scaffold method, groups used in each study including their oxygen tension, and at what stage the MSCs were exposed to hypoxia.

Study	Source of MSC	MSC Markers Assessed	Pellet Culture vs. SCAFFOLD vs. Monolayer	Oxygen Tension Used and Stage Exposed
Adesida et al. [21]	Human bone marrow	CD13, CD29, CD34, CD44, CD73, CD90, CD105, CD151	Pellet	4 groups: 1 3% O_2_ (hypoxia) during both expansion and differentiation 2 3% O_2_ (hypoxia) during expansion and 21% O_2_ (normoxia) during differentiation 3 21% O_2_ (normoxia) during expansion and 3% O_2_ (hypoxia) during differentiation 4 21% O_2_ (normoxia) during both expansion and differentiation
Baumgartner et al. [22]	Human bone marrow	CD14, CD34, CD45, CD105, CD106	Scaffold	3% O_2_ (hypoxia) vs. 21% O_2_ (normoxia) during differentiation
Boyette et al. [23]	Human bone marrow	CD34, CD45, CD73, CD90, CD105, CD146, Stro-1	Pellet	4 groups: 5% O_2_ (hypoxia) during expansion and differentiation 5% O_2_ (hypoxia) during expansion and 21% O_2_ (normoxia) during differentiation 21% O_2_ (normoxia) during expansion and 5% O_2_ (hypoxia) during differentiation 21% O_2_ (normoxia) during expansion and 21% O_2_ (normoxia) during differentiation
Cicione et al. [24]	Human bone marrow	CD29, CD34, CD44, CD45, CD73, CD90, CD105, CD106, CD66, SSEA-4, Stro-1	Pellet and then scaffold during chondrogenic differentiation	1% O_2_ (severe hypoxia) vs. 21% O_2_ (normoxia) during differentiation
Duval et al. [25]	Human bone marrow	CD34, CD45	Scaffold	5% O_2_ (hypoxia) vs. 21% O_2_ (normoxia) during differentiation
Wang et al. [26]	Human adipose tissue Human bone marrow (also had rat groups)	CD19, CD34, CD45, CD79a, HLA-DR, CD29, CD44, CD73, CD90, CD105	Pellet	1% O_2_ (hypoxia) vs. 20% O_2_ (normoxia) during the expansion of MSCs data only shown for the human adipose tissue group.
Lim et al. [27]	Human bone marrow	CD13, CD34, CD90, CD146	Bioreactor	8% O_2_ (hypoxia) vs. 20% O_2_ (normoxia) during differentiation
Khan et al., 2007 [28]	Human adipose tissue	CD13, CD29, CD34, CD44, CD90, LNGFR, Stro-1, CD56,	Pellet	5% O_2_ (hypoxia) vs. 20% O_2_ (normoxia) during differentiation
Khan et al., 2010 [29]	Human bone marrow	CD13, CD34, CD44, CD56, CD90, CD105, LNGFR, Stro1,	Pellet	5% O_2_ (hypoxia) vs. 20% O_2_ (normoxia) during differentiation
Lee et al., 2018 [30]	Human bone marrow	CD14, CD29, CD34, CD44, CD45, CD73, CD105, HLA-DR	Pellet	1% O_2_ (hypoxia) vs. 21% O_2_ (normoxia) during expansion
Lee et al., 2015 [31]	Human bone marrow	CD14, CD29, CD34, CD44, CD45, CD73, CD90, CD105, Stro-1.	Pellet	1% O_2_ (hypoxia) vs. 21% O_2_ (normoxia) during expansion
Legendre et al. [32]	Human bone marrow	CD14, CD29, CD34, CD44, CD45, CD64, CD73, CD90, CD105 CD146, HLA-DR	Scaffold	3% O_2_ (hypoxia) vs. 21% O_2_ (normoxia) during differentiation
Markway et al. [33]	Human bone marrow	CD45, CD73, CD90, CD105	Pellet or micropellet	2% O_2_ (hypoxia) for both groups during expansion and then 2% O_2_ (Hypoxia) vs. 20% O_2_ (Normoxia) during differentiation
Merceron et al. [34]	Human adipose tissue	CD29, CD34, CD44, CD45, CD90, CD105	Pellets	20% O_2_ (Normoxia) vs. 5% O_2_ (hypoxia) during differentiation
Munir et al. [35]	Human adipose tissue	CD34, CD45, CD74, CD90 and CD105, HLA-DR	Pellet/micromass+ scaffold	5% O_2_ (hypoxia) vs. 21% O_2_ (normoxia) during differentiation
Neybecker et al. [36]	Human synovium	CD34, CD45, CD73, CD90, CD105	Scaffold	5% O_2_ (hypoxia) vs. 20% O_2_ (normoxia) during differentiation
Ohara et al. [37]	Human synovium	CD45, CD73, CD90, CD105, CD140b	Pellet	5% O_2_ (hypoxia) vs. 21% O_2_ (normoxia) during expansion
Pattappa et al., 2019 [38]	Human bone marrow	None assessed; referenced previous studies using the same isolation method which assessed CD19, CD34, CD44, CD45, CD73, CD90, CD105, CD166	Pellet	2% O_2_ (hypoxia) vs. 20% O_2_ (normoxia) during both expansion and differentiation
Pattappa et al., 2013 [39]	Human bone marrow	CD11b, CD19, CD34, CD44, CD45, CD73, CD90, CD105, HLA-DR.	Pellet	5% O_2_ or 2% O_2_ (hypoxia) vs. 20% O_2_ (normoxia) during both expansion and differentiation
Tian et al. [40]	Human bone marrow	CD14, CD29, CD44, CD45, CD105	Monolayer	5% O_2_ (hypoxia) vs. 21% O_2_ (normoxia) during differentiation
Van de Walle et al. [41]	Human bone marrow	CD44, CD73, CD90, CD105, CD166	Pellet	3% O_2_ (hypoxia) then 21 % O_2_ normoxia during differentiation vs. 3% O_2_ (hypoxia) continuously during differentiation
Safwani et al [42,43]	Human adipose tissue	CD14, CD19. CD34, CD45, CD73, CD90, CD105	Pellet	2% O_2_ (hypoxia) vs. 21% O_2_ (normoxia) during both expansion and differentiation (also compared with and without fetal bovine serum in the culture medium)
Weijers et al. [43]	Human adipose tissue	CD31, CD34, CD54, CD90, CD105, and CD166	Pellet	1% O_2_ (hypoxia) vs. 20% O_2_ (normoxia) during both expansion and differentiation
Zhao et al. [44]	Human adipose tissue	CD14, CD19, CD34, CD45, CD73, CD90, CD105	Monolayer	2% O_2_ (hypoxia) vs. 21% O_2_ (normoxia) during the expansion of MSCs

**Table 2 ijms-23-15210-t002:** Study results. Shows the following information for each study included in this review- outcome measures used in each study and the results of the effect of hypoxia on chondrogenesis.

Study	Outcome Measures	Effect of Hypoxia on Chondrogenesis
Adesida et al. [21]	Safranin-O staining for sulfated glycosaminoglycan (GAG) matrix deposition sulfated GAG per DNA content Gene expression of *ACAN*, *COL1A1*, *COL2A1*, *COL10A1*, *COMP*, *HIF-1α* and *HIF-2α*, *SOX9*, *TGFβ-RI*, and *TGFβ-RII* via real-time -polymerase chain reaction (RT-PCR).	Effect of hypoxia at expansion phase: Stronger safranin-O staining for sulfated GAG matrix in both hypoxia expansion-normoxia differentiation and hypoxia expansion-hypoxia differentiation groups compared to normoxia expansion groups. Increased GAG per DNA content in both hypoxia expansion-normoxia differentiation and hypoxia expansion groups—hypoxia differentiation compared to normoxia expansion groups. Increased expression of *ACAN*, *COL2A1*, and *SOX9* in both hypoxia expansion-normoxia differentiation and hypoxia expansion-hypoxia differentiation groups compared to normoxia expansion groups. Effect of hypoxia during the differentiation phase: Increased expression of *HIF-2α*, decreased expression of *COL10A1* in hypoxia during differentiation group compared to normoxia during differentiation group at both hypoxia/normoxia during MSC expansion No statistical difference in expression of *COL1A2* and *COMP*, in hypoxia during differentiation group compared to normoxia during differentiation group at both hypoxia/normoxia during MSC expansion. Effect on TGFβ-RI and TGFβ-RII: Hypoxia-expanded cells showed greater expression of *TGFβ-RI* but no significant change in *TGFβ-RII* compared to normoxia-expanded cells when exposed to hypoxia in differentiation. Summary: groups exposed to hypoxia during expansion showed enhanced chondrogenic potential compared to normoxia exposed groups.
Baumgartner et al. [22]	*COL2A* expression via real-time PCR Alcian blue staining	Effect of hypoxia during the differentiation phase: Greater *COL2A* expression and Alcian blue staining indicate increased chondrogenic differentiation. Summary: group exposed to hypoxia during the differentiation phase showed enhanced chondrogenic differentiation compared to the normoxia exposed group.
Boyette et al. [23]	Wet pellet size Alcian Blue staining Safranin-O/Fast Green staining Sulfated GAG	Effect of hypoxia during the expansion phase compared to normoxia during the expansion phase (with hypoxia differentiation in both groups) Inhibited chondrogenesis indicated by reduced sulphated GAG and lower cartilage quality scores on histological analysis. Effect of hypoxia during the differentiation phase compared to normoxia during the differentiation phase (with normoxia expansion in both groups) Increased chondrogenesis indicated by greater wet pellet size, Alcian blue staining, and safranin-O/fast green staining. Summary: chondrogenesis is best when there is normoxia during the expansion phase followed by hypoxia during the differentiation phase
Cicione et al. [24]	Alcian blue staining Toluidine blue staining Safranin-O staining Masson’s trichome Haematoxylin and eosin staining Immunohistochemistry assessing type I collagen, type II collagen, and aggrecan. *SOX9*, *ACAN*, *COL1A1* expression via RT-PCR.	Effect of hypoxia during the differentiation phase: 1 Inhibited chondrogenesis compared to normoxic conditions in staining, and immunohistochemistry for type II collagen. (data not shown for aggrecan and type I collagen). 2 Greater *SOX9*, *ACAN*, and *COL1A1* expression in normoxia, which is then reduced in severe hypoxia. Summary: group exposed to severe hypoxia during the differentiation phase showed inhibited chondrogenic differentiation compared to normoxia-exposed group.
Duval et al. [25]	Alcain blue staining *SOX5*, *SOX6*, *SOX9*, *ACAN*, *COL2A1* expression via RT-PCR Procollagen II protein expression via Western blot analysis	Effect of hypoxia during the differentiation phase: Increased *SOX*, *ACAN*, *COL2A1* gene expression, procollagen II protein expression, and increased alcain blue staining in hypoxia exposed cells compared to normoxia exposed. Summary: groups exposed to hypoxia during the differentiation phase showed enhanced chondrogenic markers compared to the normoxia exposed group.
Wang et al. [26]	*SOX9*, *COL2A1*, *COL10A1* expression via RT-PCR Alcian blue staining Immunohistochemistry assessing type II collagen and type X collagen.	Adipose-derived stem cells: Effect of hypoxia during the expansion phase: Greater expression of *SOX9* at day 7 (but not day 21), *COL2A1* at day 21, and *COL10A1* at day 21 in hypoxia exposed group compared to the normoxia exposed group. Greater GAG content in the hypoxia exposed group compared to the normoxia exposed group as shown by Alcian Blue staining intensities. Greater expression of type II collagen and type X collagen in hypoxia exposed group than normoxia exposed group using immunohistochemistry. Summary: group exposed to hypoxia during the differentiation phase showed enhanced chondrogenic differentiation compared to the normoxia exposed group.
Lim et al. [27]	*ACAN*, *COL1A2*, *COMP*, *TGF-βR1*, *BNIP3* and *Glut1* gene expression via RT-PCR.	Effect of hypoxia during the differentiation phase: Greater expression of *ACAN*, *COL1A2*, *COMP*, *TGF-βR1*, *BNIP3* and *Glut1* in hypoxic exposed group compared to normoxia-exposed group. Summary: group exposed to hypoxia during the differentiation phase showed enhanced chondrogenic differentiation compared to the normoxia exposed group.
Khan et al., 2007 [28]	Pellet wet weight Glycosoaminoglycan per DNA accumulation Gene expression of *COL1A2*, *COL2A1*, *COL9A1*, *COL10A1*, *COL11A2*, *SOX5*, *SOX6*, *SOX9*, *HIF1α*, *HIF2α*, *ACAN*	Effect of hypoxia during the differentiation phase: Greater expression of *SOX5*, *SOX6*, *SOX9*, *ACAN*, *COL2A1*, *COL9A1*, *COL10A1*, *and COL11A2* genes in hypoxia exposed group compared to the normoxia exposed group. No difference in the *COL1A2* gene. Hypoxia exposed group showed increased levels expressed of *HIF2α*, not *HIF1α*. Hypoxia exposed group had more Glycosaminoglycan per DNA accumulation than normoxia exposed group. Hypoxia exposed group had greater pellet wet weight than normoxia exposed group Summary: group exposed to hypoxia during the differentiation phase showed enhanced chondrogenic differentiation compared to the normoxia exposed group.
Khan et al., 2010 [29]	Pellet wet weight Glycosoaminoglycan per DNA accumulation Gene expression of *COL1A2*, *COL2A1*, *COL9A1*, *COL10A1*, *COL11A2*, *SOX5*, *SOX6*, *SOX9*, *HIF1α*, *HIF2α*, *ACAN*, *VCAN*, *HSPG2*	Effect of hypoxia during the differentiation phase: Hypoxia exposed group had greater pellet wet weight than normoxia-exposed cells Hypoxia exposed group had more Glycosaminoglycan per DNA accumulation than normoxia-exposed cells. Hypoxia exposed group showed increased expression of *SOX6*, *COL2A1*, *COL11A2*, *ACAN* compared to normoxia exposed group. No difference in *COL1A2* gene expression Hypoxia exposed group did not show statistically significant differences in the expression of *HIF-1α* and *HIF-2α* compared to the normoxia exposed group. Summary: group exposed to hypoxia during the differentiation phase showed enhanced chondrogenic differentiation compared to the normoxia exposed group.
Lee et al. [30]	Toluidine blue staining Gene expression of *SOX9* via RT-PCR. Gene expression of *Collagen I*, *II*, *III*	Effect of hypoxia during the expansion phase: Hypoxia exposed group produced significantly more glycosaminoglycan compared to normoxia exposed cells. Hypoxia exposed group expressed more *SOX9* expression compared to normoxia Hypoxia exposed group showed significantly more expression of Collagen I, II, and III compared to normoxia-exposed cells. Summary: group exposed to hypoxia during the expansion phase showed enhanced chondrogenesis compared to the normoxia exposed group.
Lee et al. [31]	Pellet size Immunohistochemical staining for collagen type II Gene expression of *SOX9*, *COL2A1*, *ACAN*	Effect of hypoxia during the expansion phase: Hypoxia exposed group had greater pellet size compared to normoxia. Hypoxia exposed group showed increased type II collagen accumulation compared to normoxia. Hypoxia exposed group showed greater gene expression of *SOX9*, *COL2A1*, *and Aggreca*n compared to normoxia. Summary: group exposed to hypoxia during expansion showed enhanced chondrogenesis compared to the normoxia exposed group.
Legendre et al. [32]	Gene expression of *COL2A1*, *COL1A1*, *COL10A1* via RT-PCR.	Effect of hypoxia during the differentiation phase: Increased *COL2A1* when BMP-2/TGF-*β*1 added, increased *COL2A1/COL1A1* ratio, increased *COL2A1/COL10A1* ratio in hypoxia exposed group compared to normoxia exposed. Summary: group exposed to hypoxia during differentiation, with TGFB + BMP-2, showed the best chondrogenic differentiation.
Markway et al. [33]	(1) Sulfated *GAG* (2) *COL2A1*, *ACAN* and *SOX9* gene expression via RT-PCR.	Effect of hypoxia during the differentiation phase (with both groups exposed to hypoxia during the expansion phase): (1) Hypoxia exposed group during differentiation showed increased *COL2A1*, *ACAN* expression, and GAG content compared to normoxia exposed group. No difference in *SOX9* expression. Summary: group exposed to hypoxia in the differentiation phase showed enhanced chondrogenic markers compared to the normoxia exposed group.
Merceron et al. [34]	(1) Gene expression of *COL2A1*, *ACAN* via RT-PCR. (2) Alcian blue staining (3) Immunohistochemical staining for type II collagen	Effect of hypoxia during the differentiation phase: (1) No significant difference in *COL2A1* and *ACAN* expression between the normoxia and hypoxia exposed groups. (2) *COL2A1* expression in hypoxia exposed group was shown 7 days earlier than in the normoxic exposed groups (3) Alcian blue staining was positive with both hypoxia and normoxia exposed groups. Type II Collagen immunostaining was only highly positive under hypoxia. Summary: group exposed to hypoxia during differentiation showed upregulation of some chondrogenic markers compared to normoxia exposed group.
Munir et al. [35]	(1) Gene expression of ACAN, SOX9, COL1A1, COL2A1, COL10A1 via RT-PCR. (2) Histological evaluation with H&E and toluidine blue staining	Effect of hypoxia during the differentiation phase:Pellet culture: (1) Increased expression of *SOX9* and *COL1A1 i*n hypoxia exposed group compared to the normoxia exposed group. No significant difference in the expression of *COL2A1* and *ACAN* in hypoxia exposed group. (2) H&E staining showed more cells peripherally in hypoxia exposed group compared to in normoxia exposed. Toluidine blue stained sections showed increased matrix deposition peripherally in the hypoxic exposed group compared to the normoxia exposed group, where there was increased matrix deposition centrally. Scaffold culture: hypoxia exposed group showed increased levels of *SOX9*, and higher ratios of collagen II:I and II:X compared to normoxia. No significant differences between hypoxia and normoxia exposed groups in aggrecan, collagen I, collagen II, and collagen X gene expression. Summary: group exposed to hypoxia during the differentiation phase showed increases in some chondrogenic markers compared to the normoxia exposed group. (Some markers however showed no difference)
Neybecker et al. [36]	(1) *COL2A1*, *COL2B*, *ACAN*, *SOX9* gene expression via RT-PCR. (2) Total GAG content measured by colorimetric assay. (3) Alcain blue staining (4) Immunohistochemical staining for type II collagen	Effect of hypoxia during the differentiation phase: (1) Increased gene expression of *ACAN*, *SOX9*, *COL2A1* in hypoxia exposed group compared to normoxia-exposed group (2) No difference in GAG content between hypoxia and normoxia exposed group (3) No difference in Alcain blue staining between the hypoxia and normoxia exposed group. (4) No difference in immunohistochemical staining for type II collagen between the hypoxia and normoxia exposed groups. Summary: group exposed to hypoxia during differentiation showed enhanced chondrogenic marker gene expression but no difference at the protein level compared to the normoxia exposed group.
Ohara et al. [37]	(1) Pellet weight (2) Safranin-O staining and then Bern score	Effect of hypoxia during the expansion phase: (1) No difference in pellet weight between normoxia and hypoxic exposed group (2) No difference in Bern score after safranin-O staining between the normoxia and hypoxia exposed groups. Summary: group exposed to hypoxia during the expansion phase showed no difference in chondrogenesis compared to the normoxia exposed group.
Pattappa et al., 2019 [38]	(1) Total GAG content measured by 1,2-dimethylmethylene blue assay (2) ELISA for Collagen I and II (3) Immunohistochemistry for Collagen II (4) *SOX5*, *SOX6*, *SOX9*, *ACAN*, *LECT1*, *COL1A1*, *COL2A1*, *COL6A1*, *COL9A1*, *COL10A1* (amongst others) gene expression via RT-PCR	Effect of continuous hypoxia during expansion and differentiation phases: Donors were found to either be responsive or nonresponsive to hypoxic conditions: (1) Responsive donors: increase in collagen II content, GAG content, and wet weight in hypoxia exposed group compared to normoxia exposed group (2) Unresponsive donors: no significant difference in GAG, collagen II content, and wet weight between hypoxia exposed and normoxia groups. (3) Responsive donors: Hypoxia-exposed cells showed increased *SOX5*,*6*,*9*, *ACAN*, *COL2A1* expression Summary: there are responsive and non-responsive donors to hypoxia: in the responsive donors, the hypoxia exposed group showed enhanced chondrogenesis compared to the normoxia exposed group.
Pattappa et al., 2013 [39]	(1) Total GAG content measured by 1,2-dimethylmethylene blue assay	Effect of continuous hypoxia during expansion and differentiation phases: (1) No difference in GAG/DNA ratio between normoxia and hypoxia-exposed groups. Summary: groups exposed to either continuous hypoxia or normoxia in both the expansion and differentiation phase showed no significant difference in chondrogenesis as measured by GAG content
Tian et al. [40]	(1) Gene expression of *COL2A1* and *ACAN* via RT-PCR. (2) Immunofluorescent staining for aggrecan and type II collagen	Effect of hypoxia during the differentiation phase: (1) Increased gene expression of *COL2A1*, *ACAN* in hypoxia exposed group compared to normoxia- exposed group (2) Increased GAG content per dry weight in hypoxia exposed group than the normoxia exposed group. Total collagen content showed no significant difference. (3) Increased expression of type II collagen and aggrecan in hypoxia exposed group compared to the normoxia exposed group as shown by immunofluorescent staining Summary: group exposed to hypoxia during the differentiation phase showed enhanced chondrogenesis compared to the normoxia exposed group.
Van de Walle et al. [41]	(1) Gene expression of *COL2A1* and *ACAN* via RT-PCR.	Effect of hypoxia during the differentiation phase: (1) Increased gene expression of *COL2A1* and *ACAN* in the group exposed to normoxia then hypoxia in differentiation compared to the group exposed to hypoxia continuously. (2) Ratio of collagen II-collagen X is greater under hypoxia compared to normoxia. Summary: groups exposed to normoxia then hypoxia during differentiation showed enhanced chondrogenesis compared to hypoxia continuously.
Safwani et al. [42]	(1) Gene expression of *ACAN*, *COL2A1* via RT-PCR. (2) Alcain Blue staining	Effect of continuous hypoxia during expansion and differentiation phases: (1) Increased expression of chondrogenic genes *ACAN* and *COL2A1* in normoxia without fetal bovine serum group compared to other groups including hypoxia without fetal bovine serum and normoxia with fetal bovine serum. (2) Higher percentage of the Alcian blue staining group exposed to normoxia without fetal bovine serum, compared to those cultured in hypoxia without fetal bovine serum and normoxia with fetal bovine serum. Summary: groups exposed to normoxia without FBS showed the best chondrogenesis compared to other groups including hypoxia without FBS and normoxia with FBS.
Weijers et al. [43]	(1) Gene expression of *COL2A1* and *SOX9* via RT-PCR. (2) Alcain blue staining	Effect of continuous hypoxia during expansion and differentiation phases: (1) Hypoxia exposed group showed increased expression of *SOX9* and *COL2A1* compared to the normoxia exposed group. (2) Hypoxia exposed group showed more proteoglycan staining than the normoxia exposed group. Summary: group exposed to hypoxia during both expansion and differentiation showed enhanced chondrogenesis compared to the normoxia exposed group.
Zhao et al. [44]	(1) Gene expression of *COL2A1* and *SOX9* via RT-PCR.	Effect of hypoxia during the expansion phase: (1) Hypoxia exposed group showed less *SOX9* expression compared to the normoxia-exposed group. Summary: group exposed to normoxia showed enhanced expression of chondrogenic genes compared to hypoxia exposed group.

Abbreviations: *ACAN*, aggrecan; *BNIP3*, BCL2 Interacting Protein 3; *COL10A1*, collagen type X alpha 1 chain; *COL1A1*, collagen type I alpha 1 chain; *COL1A2*, collagen type I alpha 2 chain; *COL2A1*, collagen type II alpha 1 chain; *COL6A1*, collagen type VI alpha 1 chain; *COL9A1*, collagen IX alpha 1 chain; *COL11A2*, collagen XI alpha 2 chain; *COMP*, Cartilage Oligomeric Matrix Protein; DNA, Deoxyribonucleic acid; ELISA, enzyme-linked immunosorbent assay; FBS, fetal bovine serum; GAG, glycosaminoglycan; *Glut1*, Glucose transporter 1; H&E, hematoxylin and eosin; *HIF-1α*, Hypoxia-inducible factor 1-alpha; *HIF-2α*, Hypoxia-inducible factor 2-alpha; *HSPG2*, heparan sulfate proteoglycan 2; *LECT1*, leukocyte cell-derived chemotaxin 1; RT-PCR, real-time -polymerase chain reaction; *SOX5*, SRY-box transcription factor 5; *SOX6*, SRY-box transcription factor 6; *SOX9*, SRY-box transcription factor 9; *TGFβ-RI*, Transforming growth factor beta- type I receptor; *TGFβ-RII*, Transforming growth factor beta- type II receptor.

**Table 3 ijms-23-15210-t003:** MSC markers used across all studies included in this review and the frequency of studies using each marker.

Stem Cell Marker	Number of Studies Used in
CD11b	1
CD14	7
CD13	4
CD19	4
CD29	10
CD31	1
CD34	19
CD44	12
CD45	17
CD54	1
CD56	1
CD64	1
CD66	1
CD73	14
CD74	1
CD79a	1
CD90	20
CD105	20
CD106	2
CD140b	1
CD146	3
CD151	1
CD166	2
Stro1	5
SSEA-4	1
HLA-DR	5
LGNFR	2

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
