# Peer review of "Evaluating the Effect of Hypoxia on Human Adult Mesenchymal Stromal Cell Chondrogenesis In Vitro: A Systematic Review"

_ijms, 2022, doi:10.3390/ijms232315210_

Round 1

Reviewer 1 Report

Ranmuthu and colleagues have provided a comprehensive review, focused on the effect of hypoxia on human adult mesenchymal stromal cell (MSCs) chondrogenesis in vitro. 

The Authors presented the current state of knowledge in a clear and understandable way. They used appropriate research methods. They noted the limitations of the study. They presented the hypothetical clinical potential of the effect of hypoxia in MSCs in a very interesting way.

I read the article with great interest. I recommend for publication.

Author Response

Thank you for your kind comments.

Reviewer 2 Report

Oxygen is central to cell biology. It serves as a final electron acceptor in oxidative phosphorylation (and related risks for reactive oxygen species, a missing element of discussion in the current manuscript). Oxygen is a key element in cell survival both in vivo and in vitro. It is involved in metabolic function, signaling (Abdollahi et al., 2011), and in tissue homeostasis and regeneration (Mohyeldin et al., 2010; Mas-Bargues et al., 2019). Ranmuthu and co-authors compiled a revisional manuscript, aiming to offer a systematic review and expert analysis of published reports. With particular emphasis on cartilage tissue, an avascular environment characterized by low oxygen conditions. Cell cultures are often performed in air oxygen levels that correspond to 160mmHg (20-21% O2)(Carreau et al., 2011). As the blood flow regulates the tissue oxygen pressure (Ortiz-Prado et al., 2019) the less vascularized tissues, such as cartilage, receive less oxygen resulting in lower O2 (Mas-Bargues et al., 2019). Oxygen concentrations between 2%-9% (14.4–64.8 mmHg) are commonly considered the “physiologicaloxygen level in human tissues (and for such reason culture conditions in such O2 level are frequently defined as physoxic conditions) (Simon and Keith, 2008). Such hypoxic/ physioxic conditions are variable between tissues, but it is quite unlikely to vary as the authors stated between 1-10% O2 in cartilage human tissues. Venous blood is usually 6% or slightly higher. Arterial blood is almost double in O2 amount. Only lung epithelium is exposed to what-we-call normoxic conditions. In in vitro settings, historically, the O2 condition has sporadically been adjusted to such physiological oxygen conditions.

Oxygen tension has a role in influencing chondrogenic differentiation by stem cells, such as pluripotent stem cells or multipotent mesenchymal stromal cells (adipose-derived or bone marrow-derived MSCs). The last two somatic multipotent stem cells have been analyzed by the authors. The manuscript is interesting, but the scope is not novel. The approach is standard but contains several limitations. Specifically:

Two paragraph sections in the introduction? Intro is quite confusing and contains info poorly related to the study

The authors repeatedly referred to hypoxic conditions during cell isolation. It is not clear which studies the authors refer to. In table 1 hypoxia is limited to in vitro conditions. I was not able to find any specific description or critical analysis of cell manipulation and isolation under low oxygen conditions. Please confirm.

Collagen content is critical in healthy cartilage. The pathogenesis of cartilage pathology and arthritis is frequently uncertain but multiple factors are thought to be involved such as collagen disruption, inflammation, and tendon cells response  Collagen isoforms and analysis should be discussed and motivated rather than simply reported in Table 2

Physiological oxygen levels in BM or adipose tissue are different. Reflecting on the oxygen level in in vitro conditions. In Table 2 one study only has been reported using 8% O2. Table 2 should be rather discussed in the discussion

What is the meaning of section 2.4? Such a section sounds quite redundant with Table 1. What is the reason in distinguish 20% and 21% oxygen for normoxic conditions? Are the authors implying that such difference generates different results?

Elaborate more on section 2.5. the authors should critically discuss different results in hypoxic conditions for chondrogenic induction or expansion phase. Currently, the authors limit their observations to reporting experimental conditions like in Table 1, without elaborating or analyzing

Particularly, the authors should critically discuss results in terms of transcriptomic changes or proteomic analysis

In section 2.7 the authors admitted as previous reports highlighted “mixed conclusions

Section 2.8 has several similar considerations in common with previous sections

Interestingly, the authors correctly pointed out as current MSC products consist of a heterogenic population, not necessarily related to differences in donors. Such concepts are further implied in section 3.4. Table 3 and section 3.4 are also extremely interesting. However, no description or analysis of the relevance of different markers for chondro treatment has been offered. Please revise and elaborate

Few comments in relation to chondrogenic differentiation of MSC isolated from bone marrow or adipose tissue. Furthermore, what about other sources of MSC for chondro therapies?

Section 3.3: the authors briefly mentioned another critical parameter, serum supplementation. Animal components vs human serum replacement should also be analyzed in terms of chondro differentiation efficacy (in hypoxic or normoxic conditions)

Several sections contain redundant material. The authors keep repeating the importance of different culture conditions (oxygen level) or medium supplementations and growth factors. However, they never discuss critically or analyze such differences and effects in efficient chondrocyte maturation or functions

Cellular response to oxygen levels depends on Hypoxia-Inducible transcription Factors (HIFs) that are able to modulate more than 70 genes involved in the cells cycle, proliferation, metabolism, stem cells plasticity, angiogenesis, and immunomodulation (Majmundar et al., 2010). HIF is a heterodimeric transcription factor formed by two subunits, HIF-α and HIF-β (Muz et al., 2009). HIF-1 and -2 (alpha and beta) relevance and mechanism of action have been superficially described and seldomly put in relation to chondrocyte generation. The authors are strongly encouraged to revise and implement such a section

The supplementary table makes no sense. Most of the topics are classified as n/a or +/++; thus, a simple description and discussion in the manuscript would have been sufficient. Critical analysis of + and ++ classified studies should be implemented.

Reference editing should be revised and edited according to instructions.

Minor details:

Table 1, study nr.2 reports 106 rather than CD106

Table 1 sometimes oxygen or O2 or nothing is used before “(hypoxia)”

Ref nr. 58 is wrongly formatted

Reviewer 3 Report

In the present systematic review, Ranmuthu et al. explore the effect of hypoxia on the chondrogenesis of adult human mesenchymal stromal cells in vitro. The manuscript is well structured and analyzed 24 scientific papers on this topic. The review is interesting, and the tables make it easier and more understandable to compare all data. In my opinion the work is suitable for publication in IJMS.

Author Response

Thank you for your kind comments.

Round 2

Reviewer 2 Report

The manuscript would benefit from additional revision in English form. Several acronyms are not properly described at the first use (ie., page 2 with MMP and more)

MSCs are not the sole cells previously tested in hypoxic conditions. The reason behind such alternative cultural conditions is not properly described

The authors continuously use MSC or “mesenchymal stem cells”. Once the acronym has been introduced, the authors are called to strictly refer to such definition for all the pages down, to avoid confusion and redundancy.

The discussion states initially “There were several limitations to this study.”. This is not an original study. The manuscript is an overview of selected previous reports.

We respect authors’ position and explanation. Nevertheless, a critical discussion on the effects generated by oxygen levels in ex vivo conditions on chondrocyte maturation or functions cannot be left out from an overview manuscript like this.

Finally, we would like to remind the authors as MSC is the acronym for mesenchymal stromal cells. More than 30 years ago, Dr. Caplan provocatively coined the term “mesenchymal stem cells”, in response to their multipotency and highly proliferative capacity (Caplan, A.I. J. Orthop. Res. 1991). However, several years later, he publicly admitted the importance to refine such multipotent cells as stromal medicinal products, whose paracrine action rather than differentiation capacity leads to regeneration induction. The exact mechanism(s) of action for different MSC is still largely unknown, but it is well accepted as they home in sites of injury or paracrinely induce regenerative effects through secreted bioactive factors and trophic mediators. Indeed, leading experts have highlighted that patient’s own tissue-resident progenitor cells are the real fabricator for new tissue, supported and enhanced by MSC-secreted bioactive factors (Caplan, A.I. Stem Cells Transl. Med. 2017). The nomenclature changed officially and has been consolidated by the International Society for Cellular Therapy (ISCT) in 2019, 15 years after delineating the release criteria to identify and release MSC products (Dominici, M et al. Cytotherapy 2006). These cells have been defined as “mesenchymal” since the mesenchyme cells are defined as a type of tissue surrounded by a large extracellular matrix (ECM) and characterized by loose intercellular adhesion and lack of polarity. They have been called “stromal” cells as well since, as any structural component in connective tissue, fibroblastic cells do adhere to culture-treated plastic and expand massively.

Their affiliation within the “stem” group does not require such “degree” in the name

Author Response

Reviewer’s Comment

Author’s Revision 

The manuscript would benefit from additional revision in English form. Several acronyms are not properly described at the first use (ie., page 2 with MMP and more)

Many thanks to the reviewer for this comment. Acronyms have now been described at first use.  

MSCs are not the sole cells previously tested in hypoxic conditions. The reason behind such alternative cultural conditions is not properly described

Many thanks for this comments - have emphasised in the conclusion that there are a number of key players in cartilage regeneration, and the effects of hypoxia on the various cell types (ranging from progenitor cells to the differentiated cell) should be investigated in more detail.

The authors continuously use MSC or “mesenchymal stem cells”. Once the acronym has been introduced, the authors are called to strictly refer to such definition for all the pages down, to avoid confusion and redundancy.

Many thanks to the reviewer for this comment. MSC has been used throughout the paper to avoid confusion after the first definition of MSC as ‘mesenchymal stromal cell’.    

The discussion states initially “There were several limitations to this study.”. This is not an original study. The manuscript is an overview of selected previous reports.

Many thanks to the reviewer for this comment. The word ‘study’ has been changed to ‘review’.  

We respect authors’ position and explanation. Nevertheless, a critical discussion on the effects generated by oxygen levels in ex vivo conditions on chondrocyte maturation or functions cannot be left out from an overview manuscript like this.

Many thanks to the reviewer for this comment. A section (3.7) has been added on the effect of hypoxia on chondrocyte function. 

Finally, we would like to remind the authors as MSC is the acronym for mesenchymal stromal cells. More than 30 years ago, Dr. Caplan provocatively coined the term “mesenchymal stem cells”, in response to their multipotency and highly proliferative capacity (Caplan, A.I. J. Orthop. Res. 1991). However, several years later, he publicly admitted the importance to refine such multipotent cells as stromal medicinal products, whose paracrine action rather than differentiation capacity leads to regeneration induction. The exact mechanism(s) of action for different MSC is still largely unknown, but it is well accepted as they home in sites of injury or paracrinely induce regenerative effects through secreted bioactive factors and trophic mediators. Indeed, leading experts have highlighted that patient’s own tissue-resident progenitor cells are the real fabricator for new tissue, supported and enhanced by MSC-secreted bioactive factors (Caplan, A.I. Stem Cells Transl. Med. 2017). The nomenclature changed officially and has been consolidated by the International Society for Cellular Therapy (ISCT) in 2019, 15 years after delineating the release criteria to identify and release MSC products (Dominici, M et al. Cytotherapy 2006). These cells have been defined as “mesenchymal” since the mesenchyme cells are defined as a type of tissue surrounded by a large extracellular matrix (ECM) and characterized by loose intercellular adhesion and lack of polarity. They have been called “stromal” cells as well since, as any structural component in connective tissue, fibroblastic cells do adhere to culture-treated plastic and expand massively.

Their affiliation within the “stem” group does not require such “degree” in the name

Many thanks to the reviewer for this comment. The term ‘mesenchymal stem cell’ has been removed from the manuscript and ‘mesenchymal stromal cell’ has been used.